# Review of the Hydrogen Permeation Test of the Polymer Liner Material of Type IV On-Board Hydrogen Storage Cylinders

**DOI:** 10.3390/ma16155366

**Published:** 2023-07-31

**Authors:** Xiang Li, Qianghua Huang, Yitao Liu, Baodi Zhao, Jiepu Li

**Affiliations:** 1China Special Equipment Inspection and Research Institute, Beijing 100029, China; lixiang@csei.org.cn (X.L.); qhhamstd@sina.com (Q.H.); liuyitao@csei.org.cn (Y.L.); zhaobaodi198800@163.com (B.Z.); 2Key Laboratory of Safety of Hydrogen Energy Storage and Transportation Equipment for State Market Regulation, Beijing 100029, China

**Keywords:** hydrogen storage cylinder, liner material, polymer, hydrogen permeation test, standard

## Abstract

Type IV hydrogen storage cylinders comprise a polymer liner and offer advantages such as lightweight construction, high hydrogen storage density, and good fatigue performance. However, they are also characterized by higher hydrogen permeability. Consequently, it is crucial for the polymer liner material to exhibit excellent resistance to hydrogen permeation. International organizations have established relevant standards mandating hydrogen permeation tests for the liner material of type IV on-board hydrogen storage cylinders. This paper provides a comprehensive review of existing research on hydrogen permeability and the hydrogen permeation test methods for the polymer liner material of type IV on-board hydrogen storage cylinders. By delving into the hydrogen permeation mechanism, a better understanding can be gained, offering valuable references for subsequent researchers in this field. This paper starts by thoroughly discussing the hydrogen permeation mechanism of the liner material. It then proceeds to compare and analyze the hydrogen permeation test methods specified by various standards. These comparisons encompass sample preparation, sample pretreatment, test device, test temperature and pressure, and qualification indicators. Then, this study offers recommendations aimed at enhancing the hydrogen permeation test method for the liner material. Additionally, the influence of test temperature, test pressure, and polymer material properties on the hydrogen permeability of the liner material is discussed. Finally, the influences of the test temperature, test pressure, and polymer material properties on the hydrogen permeability of the liner material are discussed. Future research direction on the hydrogen permeability and hydrogen permeation test method of the liner material of the type IV hydrogen storage cylinder has been prospected.

## 1. Introduction

With the increasingly serious global greenhouse effect, the goals of carbon emissions peak and carbon neutrality have been proposed, and various countries have accelerated their exploration of new energy sources [1,2,3]. Hydrogen energy is considered the most promising secondary energy source of this century due to its advantages of abundant reserves, pollution-free operations, high combustion calorific value, zero emissions, and renewability [4,5]. It represents the most effective technical approach to achieving the goals of carbon emission peak and carbon neutrality [6,7,8]. In the field of transportation, hydrogen fuel cell vehicles (HFCVs) are the main applications of hydrogen energy and have attracted substantial attention to related technologies [9,10]. Currently, the most mature and widely used approach to on-board hydrogen storage is high-pressure gaseous hydrogen storage, primarily utilizing the on-board type III and type IV hydrogen storage cylinders with nominal working pressures (NWP) of 35–70 MPa [11,12]. Among HFCVs, the type IV hydrogen storage cylinder has become a research hotspot due to its advantages of being lightweight, having high hydrogen storage density, and good fatigue performance [13,14]. The structure of the type IV hydrogen storage cylinder and the polymer liner are shown in Figure 1. Typically, the polymer liner material of type IV hydrogen storage cylinders is made from high-density polyethylene (HDPE) or polyamide (PA) [15,16,17]. However, due to the structural characteristics of these polymer materials, hydrogen permeation is inevitable, and the large gas permeability and dissolution rate can also lead to liner blistering and collapse [18,19].

The hydrogen explosion limit is 4–75.6%. When a hydrogen fuel cell vehicle is parked in a relatively hermetic space for a long time, hydrogen can accumulate, increasing the risk of fire and explosion [20,21]. To ensure the safety of on-board hydrogen storage cylinders, the ISO 19881 standard stipulates that the steady-state hydrogen permeability rate of a type IV hydrogen storage cylinder should be less than 46 NmL/(h·L) at 1.15 times the nominal working pressure (NWP) and 55 °C, and less than 6 NmL/(h·L) at the NWP and 15 °C [22]. Therefore, the hydrogen permeability of the liner material is critical in determining the intrinsic safety of the type IV hydrogen storage cylinder. Reducing the hydrogen permeation amount of liner materials as much as possible and accurately measuring the hydrogen permeability rate remain essential and challenging tasks that will be the focus of future research.

Various countries and organizations have issued standards that define hydrogen compatibility test methods (including hydrogen permeation tests) for the polymer liner materials used within the on-board hydrogen storage cylinders. The CSA/ANSI CHMC 2:19-2019 standard “Test methods for evaluating material compatibility in compressed hydrogen applications—Polymers” (abbreviated as CSA/ANSI CHMC 2) [23] was revised by the Canadian Standards Association and CSA America in 2019. This important hydrogen compatibility test requires that hydrogen permeation testing of the liner material use the high-pressure hydrogen gas permeation test (HPHP) method. In the HPHP method, the sample is placed between a high-pressure chamber and a low-pressure chamber, and the gas permeability coefficient is determined by the extent of gas permeation between the two chambers. The change in the amount of hydrogen permeating the sample over time is recorded when the hydrogen pressure in the high-pressure chamber reaches a steady state. This test method is applicable to compressed hydrogen applications and is not limited to the on-board high-pressure hydrogen storage systems. The ISO 11114-5-2022 standard “Gas cylinders—Compatibility of cylinder and valve materials with gas contents—Part 5: Test methods for evaluating plastic liner” (abbreviated as ISO 11114-5) [24] was revised by the International Organization for Standardization in 2022. Two types of tests are defined in this standard: (a) permeation tests on discs or other specimens, which are used to compare different polymer materials and select the right liner material or to identify areas of the liner that might undergo excessive hydrogen permeation; and (b) cylinder permeation tests, which are used to determine liner material suitability. The hydrogen permeation test method in ISO 11114-5 also uses the HPHP method. China has not issued a separate national standard for hydrogen compatibility testing of polymer liner materials. As a normative appendix to the test standard for hydrogen storage cylinders, the China Association for Technical Supervision Information revised group standard T/CATSI 02 007-2020 “Fully-wrapped carbon fiber reinforced cylinder with a plastic liner for on-board storage of compressed hydrogen for land vehicles” (abbreviated as T/CATSI 02 007) [25] was released in 2020. This standard stipulates that hydrogen permeation tests of liner materials for type IV on-board hydrogen storage cylinders use the HPHP method. There are differences in the hydrogen permeation test methods of polymer liner materials in each standard, and no unified measurement method has been formed. Furthermore, the hydrogen permeability of polymers such as HDPE and PA is affected mainly by environmental conditions (temperature and pressure), material crystallinity, molecular chains, fillers, and interaction between the gas and material [26,27]. Several scholars have researched the laws that govern the influences of various factors on polymer permeability [28,29,30]. However, there is little research on hydrogen permeation test methods of the polymer liner material of on-board type IV hydrogen storage cylinders.

Section 2 discusses the hydrogen permeation mechanism within the polymer liner material. In Section 3, the hydrogen permeation test methods of polymer liner materials required by various standards are compared and analyzed. This section considers factors such as sample preparation, sample pretreatment, test device, test temperature, pressure, and qualification indicators. Several recommendations for improving the hydrogen permeation test method of the liner material are presented. Section 4 discusses the influences of test temperature, test pressure, and polymer material properties on the hydrogen permeability of the liner material based on literature research. Section 5 provides insights into future research directions on the hydrogen permeability and hydrogen permeation test method of the liner material for type IV hydrogen storage cylinders.

## 2. Hydrogen Permeation Mechanism in Polymer Liner Materials

Hydrogen embrittlement may occur when the metal liner materials of type II and type III hydrogen storage cylinders come into contact with high-pressure hydrogen. Hydrogen atoms permeate into the metal material and combine to form hydrogen molecules [31]. As the partial hydrogen concentration in the metal liner saturates, the mechanical properties and plasticity of the metal liner decrease, leading to the potential development of cracks or delayed fracture [32,33]. On the other hand, type IV hydrogen storage cylinders use polymer materials to achieve high hydrogen storage densities. Research conducted by the Sandia National Laboratory in the US [34] demonstrates that polymers do not suffer from hydrogen embrittlement as metals do in high-pressure hydrogen environments. This is because the hydrogen absorbed by polymers exists in the form of diatomic molecules and does not segregate, unlike in metals. However, it is worth noting that HDPE and PA, which are commonly used as liner materials in type IV cylinders, are semi-crystalline thermoplastics. Hydrogen molecules permeate slowly through the amorphous zones in their polymeric molecular structures [35,36]. Gas molecule transport in such polymers occurs via a dissolution–diffusion mechanism [37,38]. These differences in hydrogen permeation mechanisms highlight the varying behaviors of metal and polymer materials in hydrogen-rich environments. The gas transport theory proposed by Calvert [39] and Klopffer M.H. [38] assumes that when the polymer comprises a homogeneous, non-porous film at a given temperature, the diffusion process of gas in the polymer can be divided into five steps, as shown in Figure 2. 

During the first stage, gas molecules diffuse through the limit layer on the upstream side of the material. In the second stage, these gas molecules are adsorbed by the polymer due to chemical affinity or solubility at the upstream interface. The third stage involves the gas molecules diffusing inside the polymer. During the fourth stage, the gas molecules are received on the downstream side. In the fifth stage, the gas molecules diffuse through the limit layer on the downstream side. The gas barrier properties of the first and fifth stages can be neglected due to the low gas-permeation barrier properties of the limit layer in these referenced stages. The formation of such limit layers is difficult to observe [35]. Therefore, the gas permeation process can be divided into two parts: dissolution and diffusion. Dissolution is a thermodynamic process that depends on the interactions between a gas and a polymer, as well as the compressibility of the gas [40,41]. When hydrogen dissolves in a polymer, it has a plasticizing effect, reducing the material’s strength and increasing its toughness [36]. On the other hand, diffusion is a kinetic term that reflects the mobility of gas molecules in a polymer due to random motion. Permeation is evaluated using two thermodynamic parameters, the diffusion coefficient (D) and the solubility coefficient (S) [42]. The diffusion coefficient (D) describes the permeability of the gas in the polymers, while the solubility coefficient (S) describes the amount of gas contained in the polymer [43].

The steady-state process of gas diffusion in a polymer conforms to Fick’s first law:(1)$$J=-D\frac{\partial c}{\partial x}$$
where *J* denotes the gas diffusion flux (mol/m^2^·s); *D* is the diffusion coefficient (m^2^/s); *c* is the molarity of hydrogen (mol/m^3^), and *x* represents the diffusion direction. 

The solubility of hydrogen in a polymer conforms to Henry’s law. The hydrogen molecule concentration is proportional to the partial pressure of hydrogen [19,36], as shown in Equation (2):(2)$$c=Sp_{H}$$
where *S* denotes the solubility coefficient (mol/m^3^·Pa), and *p_H_* is the partial pressure of hydrogen (Pa). The hydrogen permeability coefficient of a polymer is the product of the diffusion and solubility coefficients and is expressed as *P_e_* = *D* × *S* [44,45].

When hydrogen permeates a polymer of a constant thickness (*t* = *x*_1_ − *x*_2_) from the upstream side and forms a pressure difference in Δ*t_H_* on the downstream side, the hydrogen permeation flux *J* can be expressed as shown in Equation (3):(3)$$J=-D\frac{S(p_{H2}-p_{H1})}{\left(x_{1}-x_{2}\right)}=DS\frac{\Delta p_{H}}{t}=P_{e}\frac{\Delta p_{H}}{t}$$

The diffusion coefficient *D* can be obtained from the lag time *τ* via extrapolation from the linear portion, in which the hydrogen transfer rate reaches a steady state, to the time axis in the integrated transmission curve [46], as shown in Equation (4):(4)$$D=\frac{l^{2}}{6\tau }$$
where *l* is the sample thickness (m), and τ is the lag time (s).

For a circular specimen with a thickness of t (m) and a hydrogen-exposed area of *A* (m^2^), the steady-state gas permeation rate *F_g_* (mol/s) [47] is shown in Equation (5):(5)$$F_{g}=J\cdot A$$

Inserting Formula (5) into Formula (3) allows the permeability coefficient *P_e_* to be calculated as shown in Formula (6):(6)$$P_{e}=\frac{F_{g}\cdot t}{\Delta p_{H}\cdot A}$$

The standard unit of the hydrogen permeability coefficient *P_e_* of a polymer is mol·m/(m^2^·s·Pa). The hydrogen permeation coefficient of a polymer can be calculated as the product of the molar mass of the permeating hydrogen gas at permeation equilibrium and the sample thickness, divided by the hydrogen-exposed area of the sample, the pressure difference across the sample, and the time to reach steady-state permeation. 

Based on the hydrogen permeation mechanism described above for polymer liner materials, all existing standards adopt the high-pressure hydrogen gas permeation test method to measure the hydrogen permeation coefficient of liner materials. The calculation formula for the hydrogen permeability coefficient in each standard is represented by Formula (6). The differences in hydrogen permeation test requirements among these standards will be compared and analyzed in Section 3.

## 3. Hydrogen Permeation Test Methods of the Polymer Liner Material

### 3.1. Sample Preparation

The sample preparation methods required for hydrogen permeation tests CSA ANSI CHMC 2, ISO 11114-5, and T/CATSI 02 007 are shown in Table 1.

As shown in Table 1, different standards have varying requirements for hydrogen permeation samples. CSA ANSI CHMC 2 and ISO 11114-5 specify that samples should be prepared from the polymer liner. If a comparison of hydrogen permeation performance among multiple liner materials is required, samples can be prepared from polymers produced using the same molding process as the final product. However, the above two standards do not specify sampling position requirements. T/CATSI 02 007 stipulates that samples can be prepared from the polymer liner. For seamless liners, the sampling position is located in the middle of the cylinder, while for welded liners, it is located in the middle of the cylinder, away from the welding seam. Regarding the sample source, when samples are prepared from polymers produced using the same molding process as the final product, material grade and heat treatment processes, such as curing, can affect the hydrogen permeability characteristics of the polymer liner material. Additionally, the sample may have defects, such as wrinkles, creases, and gas cavities formed during molding. Differences in the molding processes used with the prepared specimen and the final product can impact the representativeness of test results [43]. Therefore, if the sample is prepared using the same process as the liner product, it is necessary to specify that the process flow and heat treatment method are consistent with that used to make the liner and that the prepared sample should not have manufacturing defects that affect test results.

In terms of sampling position, injection-molded liners with welds may have defects such as wrinkles, creases, and gas cavities near the weld, which can affect the hydrogen permeability at the welding seam of the liner. There is still a lack of systematic research on the hydrogen permeability at the welding seam of the liner material for the type IV on-board hydrogen storage cylinders. Therefore, it is necessary to study the performance at the welding seam to ensure that the plastic liner hydrogen permeation test considers the area with the weakest hydrogen permeability.

Regarding the sample diameter, CSA ANSI CHMC 2 stipulates that, in order to ensure the consistency of hydrogen permeation test results from the same materials and the comparability of the permeability of different liner materials, the sample diameter should be greater than or equal to 25 mm. A sample diameter of 78 ± 1 mm is recommended, as it matches that of the test device described in the standard. ISO 11114-5 specifies that the sample diameter should be between 40 mm and 80 mm, while T/CATSI 02 007 requires a sample diameter of 78 ± 1 mm. Since the hydrogen permeation sample is sealed in the test device using sealing rings and the support fixture, the sample diameter only needs to exceed the hydrogen leakage area diameter. The sample volume outside the sealing rings should be as small as possible to prevent hydrogen from escaping. Therefore, the sample diameter only needs to exceed that of the hydrogen exposure area of the test device. To ensure the consistency of experimental parameters, a fixed recommended sample diameter can be specified in the standard.

CSA ANSI CHMC 2 requires a sample thickness between 1 mm and 6 mm, with a preference for a sample with the same thickness as the liner. On the other hand, both ISO 11114-5 and T/CATSI 02 007 stipulate that the sample thickness should equal the liner thickness. It is important to avoid machining the sample to change its thickness, as this may create new defects on its surface, thus affecting the hydrogen permeation test results. Therefore, when conducting a hydrogen permeation test for a specific liner material, it is recommended to select the liner thickness as the sample thickness. Furthermore, Fujiwara [41] studied the relationship between the thicknesses of different samples and the diffusion coefficient delay time and verified a linear relationship between the square of the sample thickness and the delay time. This relationship conforms to Formula (4) of the diffusion coefficient. However, since the hydrogen permeability coefficient is determined by both the diffusion and solubility coefficients, whether the sample thickness has a linear relationship with the hydrogen permeability coefficient requires further verification through experimental research.

In summary, the hydrogen permeation test should consider the influence of the molding process and weld defects of the liner material and determine appropriate sample preparation requirements and sampling position. When discussing the specific reasons in depth, they may be closely related to polymer material properties such as molecular chains, crystallinity, and additives. The influence of polymer material properties on the hydrogen permeability of the liner material is discussed in Section 4.3.

### 3.2. Sample Pretreatment

The sample pretreatment requirements for CSA ANSI CHMC 2, ISO 11114-5, and T/CATSI 02 007 are shown in Table 2.

As shown in Table 2, each standard requires heated drying and vacuum degassing of samples before hydrogen permeation testing. ISO 11114-5 recommends a vacuum pressure of 10–50 mbar, whereas CSA ANSI CHMC 2 and T/CATSI 02 007 do not specify the vacuum pressure. Regarding finishing conditions, CSA ANSI CHMC 2 requires that the mass loss rate within 48 h or 1 h of drying is <0.5%, and ISO 11114-5 requires a mass loss rate within 24 h of drying of <0.1%. It is worth noting that volatile organic compounds (VOCs) from polymer exhaust can affect hydrogen compatibility evaluation test results [22]. Therefore, it is suggested that the liner hydrogen permeation test specifications clarify the required vacuum pressure and specify the mass loss rate of the sample at the end of vacuum drying to ensure that the polymer sample completes the exhaust process. 

Regarding the drying temperature, CSA ANSI CHMC 2 recommends a drying temperature of 60 °C; ISO 11114-5 recommends 65 °C, and T/CATSI 02 007 does not give a specific temperature. A study by Emanuele Parodi et al. [48] revealed that PA6, an adipose polyamide consisting of amide and carbonyl groups, exhibited strong hygroscopicity due to its polar character. When exposed to a humid environment, PA6 absorbs water up to a saturation level, resulting in the breaking of hydrogen bonds between the chains and the formation of new hydrogen bonds with the absorbed water molecules. This phenomenon leads to plasticization and depression of the glass transition temperature, resulting in considerable degradation of the mechanical properties. Figure 3 shows the X-ray diffraction (XRD) results and crystalline fractions of PA6 at different relative humidity. As illustrated, the higher the relative humidity, the lower the glass transition temperature of PA6. Under certain conditions, PA6 may experience hydration-induced crystallization at room temperature, increasing its crystallinity and reducing its gas permeability.

Currently, there are numerous studies on the effect of humidity on the crystal structure and mechanical properties of polymer materials, but there is limited research and test data on the hydrogen permeability of liner materials for on-board hydrogen storage cylinders under varying humidity conditions. If the actual working conditions of the liner material are not considered, the sample should be dried during the hydrogen permeability test. It is recommended to keep the drying temperature below the softening temperature of the liner material to prevent softening from affecting test result consistency. However, since on-board hydrogen storage cylinders may operate in high-temperature, high-pressure, and high-humidity environments, it is essential to study the comprehensive influence of humidity, coupled with hydrogen temperature and pressure, on the hydrogen permeability of the liner material to ensure that the hydrogen permeation performance of the liner material meets applicable requirements under high-humidity conditions.

### 3.3. Test Device

The test devices used in CSA ANSI CHMC 2, ISO 11114-5, and T/CATSI 02 007 are shown in Table 3.

As shown in Table 3, the hydrogen permeation device structures described in the CSA ANSI CHMC 2 and ISO 11114-5 standards are nearly identical, but T/CATSI 02 007 does not specify a test device. Each standard uses the HPHP test method. The test device in the CSA ANSI CHMC 2 and ISO 11114-5 standards mainly consists of a high-pressure sealing cavity, a low-pressure sealing cavity, sealing rings, a wire mesh, and a sintered metal plate. The high-pressure and low-pressure sealed chambers are used to hold samples, while sintered metal plates support wafer specimens to prevent the sample from being squeezed and deformed under high-pressure gas. The wire mesh is placed between the sample and the sintered metal plate to avoid direct contact between the sintered metal and the sample surface and to ensure that hydrogen permeating from the sintered metal is distributed evenly in the exposed area of the sample. 

The HPHP method is based on the differential pressure method proposed by Barrier and Brubaker et al. [49,50,51], optimized and improved by integrating the ASTM 1434 and ISO 7229 standards, which were used to measure gas permeation properties at equilibrium. In this method, the sample is placed between the high- and low-pressure chambers, and the gas permeability coefficient is determined by measuring the amount of gas that permeates to the low-pressure chamber. Yamabe et al. [52] also proposed a thermal desorption analysis (TDA) method for measuring the hydrogen permeability coefficient. The sample was exposed to high-pressure hydrogen, and then the emission (elimination) of hydrogen from the polymer material was measured using a gas chromatograph in a non-equilibrium state. The gas permeability and diffusion coefficients were calculated by function fitting. Fujiwara [53] compared hydrogen permeability data of different polymer materials measured by the TDA and HPHP methods. As shown in Figure 4, the permeability coefficients measured by the TDA method were greater than those measured by the HPHP method regardless of the material considered, and the difference between the results increased with the test pressure. The HPHP method can measure the true extent of hydrogen permeation in the high-pressure equilibrium state, and the data are more accurate and more suitable for measuring the hydrogen permeation characteristics of high-pressure hydrogen storage cylinder liner materials [54]. However, the conditions required for continuous data acquisition and pressure maintenance under high-pressure conditions are relatively harsh, the test device is relatively complex, and the test duration is longer.

Regarding the sintered metal plate porosity, CSA ANSI CHMC 2 stipulates that the sintered metal should have a porosity of Grade 2 to ensure that the gas transmission rate of the plate is at least 100 times greater than that of the liner specimen, preventing the gas barrier properties of the sintered metal from affecting test results. Both CSA ANSI CHMC 2 and ISO 11114-5 recommend placing a 140 μm and 150 μm wire mesh, respectively, between the polymer specimen and the porous plate. Since the gas transmission rate of the wire mesh is significantly higher than that of the polymers, the mesh size has no effect on the hydrogen permeation test results [55]. Regarding the hydrogen leakage zone diameter, CSA ANSI CHMC 2 and T/CATSI 02 007 stipulate that it should be greater than or equal to 25 mm, while ISO 11114-5 requires it to be greater than or equal to 30 mm. Theoretically, the amount of hydrogen permeation is proportional to the hydrogen exposure zone diameter, and different hydrogen permeation areas should not affect the hydrogen permeation coefficient. However, some studies have shown that the measured gas permeability coefficient of a sample may appear high when the hydrogen permeation area is small. This could be due to non-isotropic hydrogen diffusion in the thickness direction of the liner [46]. Therefore, further research is needed to understand the influence of the hydrogen exposure zone diameter on hydrogen permeability to determine the appropriate diameter range.

In summary, the three standards define test devices with similar principles and structures, and all use the HPHP method. However, this method requires a higher sealing effect for the O rings. The effect of the O ring hydrogen permeability coefficient cannot be accurately judged due to the existing test device being sealed via sealing rings. Additionally, since the sample diameter is larger than that of the sealing rings, hydrogen may escape from the sample edge. Hence, it is necessary to optimize the structure of the existing test device to improve hydrogen permeation test accuracy.

### 3.4. Test Temperature and Pressure

The CSA ANSI CHMC 2, ISO 11114-5, and T/CATSI 02 007 temperatures and pressures are shown in Table 4.

As shown in Table 4, ISO11114-5 stipulates that the hydrogen permeation test should be carried out at a certain pressure and temperature, but these values are not specified. However, the test temperature and pressure specifications differ for the other two standards. CSA ANSI CHMC 2 requires the sample to undergo a permeation test at pressures of 0.1 NWP and 1.25 NWP, as well as at test temperatures of 85 °C and 15 ± 5 °C. T/CATSI 02 007 specifies permeation tests at pressures of 0.1 MPa and 1.15 NWP, and at maximum ambient temperatures of 55 ± 1 °C and 15 ± 5 °C.

The choice of test temperature should align with the working conditions and service environment. A maximum allowable temperature of 85 °C is reached temporarily during the filling of on-board hydrogen storage cylinders. Since hydrogen refueling stations are open spaces where hydrogen does not accumulate easily, temperatures above 50 °C can persist for weeks in desert areas [55]. However, in confined spaces such as garages, where hydrogen fuel cell vehicles may be parked for extended periods, hydrogen permeation into the space can cause hydrogen accumulation, leading to potential hazards. Therefore, a high-temperature hydrogen permeation test temperature of 55 °C seems reasonable.

Hydrogen storage systems utilize the SAEJ 2601 filling protocol, where the maximum filling density (100% SOC) of the on-board hydrogen storage system is achieved at a hydrogen temperature of 85 °C and a filling pressure of 1.25 NWP. Conversely, the filling pressure required to reach 100% SOC is NWP at 15 °C and 1.15 NWP at 55 °C [56]. Therefore, a high-temperature, high-pressure hydrogen permeation test conducted at 55 °C and 1.15 NWP is more representative of actual working conditions. CSA ANSI CHMC 2 specifies a pressure of 1.25 NWP for normal-temperature, high-pressure hydrogen permeation tests. Considering that cylinders are used in a normal-temperature environment where gas pressure does not exceed the nominal working pressure, it is more reasonable to perform these tests at 15 °C and the nominal working pressure. Additionally, gas cylinders may be in a no-load state with a slight positive pressure for extended periods. Therefore, it is more reasonable to perform normal-temperature, high-pressure hydrogen permeation tests at 15 °C and the nominal working pressure. In addition, since the gas cylinder may be in a no-load state with a slight positive pressure for a long time, it is more reasonable to use a no-load pressure of 0.1 MPa during a hydrogen permeation test in order to verify the hydrogen permeability of the liner material in a no-load state. 

In summary, none of the standards consider the hydrogen permeability of the polymer liner material in a low-temperature environment, and whether the combination of test pressure and test temperature is reasonable remains to be studied. The influence of temperature and pressure on the hydrogen permeability of the liner material is discussed in Section 4.1 and Section 4.2.

### 3.5. Qualification Indicators

The CSA ANSI CHMC 2, ISO 11114-5, and T/CATSI 02 007 standard hydrogen permeation test qualification indicators are shown in Table 5.

As shown in Table 5, the ISO 11114-5 standard does not specify hydrogen permeation test qualification indicators. However, for type-IV hydrogen storage cylinders, the steady-state hydrogen permeability rate should meet certain criteria. It should be less than 46 NmL/(h·L) at 1.15 NWP and 55 °C and less than 6 NmL/(h·L) at NWP and 15 °C [21,55,57]. T/CATSI 02 007 assigns a maximum allowable hydrogen permeability coefficient value range to each gas cylinder volume range. The hydrogen permeability coefficient of the polymer liner material should not exceed 2 × 10^−13^ cm^3^·cm/(cm^2^·s·Pa) at 15 °C or 1 × 10^−12^ cm^3^·cm/(cm^2^·s·Pa) at 55 °C. Based on a requirement of 6 NmL/(h·L) for a 60 L cylinder, the CSA ANSI CHMC 2 standard uses a different approach by stipulating that the steady-state gas transmission rate measured at 15 °C for a sample with a diameter of 78 mm be divided into six rating values. Grade 0 indicates the material with the worst hydrogen barrier properties, and grade 10 represents the material with the best hydrogen barrier properties. This quantitative classification of the hydrogen transmission rate provides a standardized system for comparing the hydrogen permeation performances of different materials. Based on specific hydrogen system application scenarios, the minimum acceptable hydrogen permeation test rating value can be specified, providing a clear range of hydrogen permeability coefficient values. This allows for a more accurate evaluation of material suitability for hydrogen storage applications. 

In summary, the quantitative classification of the hydrogen transmission rate provides a valuable means to compare the hydrogen permeation performances of various polymer materials. The minimum acceptable hydrogen permeation test rating value can be adjusted based on the specific requirements of different compressed hydrogen systems. However, when using the hydrogen permeation test method to assess whether a chosen liner material possesses good hydrogen permeation resistance, it is advisable to establish a maximum allowable hydrogen permeability coefficient range for the liner material. This can be determined by considering the hydrogen permeation test indicators of gas cylinders and the volume range of gas cylinders. Setting such limits ensures that the selected liner material meets the necessary hydrogen barrier properties for practical applications in different gas storage systems.

## 4. Discussion

The influence of test temperature, test pressure, and polymer material properties on the hydrogen permeability of the liner material is discussed in this section. 

### 4.1. Test Temperature

According to the GTR 13 and ISO 19881 standards, the hydrogen temperature in the on-board hydrogen storage system cannot exceed 85 °C during filling at the highest filling pressure [57,58]. The hydrogen must be pre-cooled to −40 °C before filling to avoid over-temperature during the filling process [56]. Therefore, the working temperature range of a hydrogen storage cylinder is −40 °C to 85 °C, and the influence of temperature change on the hydrogen permeability of the polymer liner material is an important research question.

Early theoretical studies [59,60,61] showed that the effects of the temperature on the polymer permeability, diffusion, and solubility coefficients conform to Arrhenius’ law within a certain temperature range:(7)$$P_{e}(T)=P_{e0}\exp(-\frac{E_{p}}{RT})$$
(8)$$D(T)=D_{0}\exp(-\frac{E_{D}}{RT})$$
(9)$$S(T)=S_{0}\exp(-\frac{\Delta H_{s}}{RT})$$
where *P_e_*_0_, *D*_0_, and *S*_0_ represent the limit values of various transportation coefficients when the temperature tends to infinity. *E_P_*, *E_D_*, and Δ*H_S_*, respectively, represent the apparent activation energies of the permeation and diffusion processes, and the heat of dissolution required for the penetrant to dissolve in the polymer matrix. The relationship between *E_P_* = Δ*H_S_* × *E_D._* shows that for hydrogen, both Δ*H_S_* and *E_D_* are positive [41]. Therefore, at a given pressure, the hydrogen permeation coefficient of the material increases as the temperature increases.

PA6 and HDPE are the most common type-IV on-board hydrogen storage cylinder liner materials [62,63]. Barth et al. [34] analyzed the temperature dependence of hydrogen permeability of several polymer materials (such as HDPE, PA, PVC, butyl rubber, etc.). The permeation of hydrogen in polymer materials at different temperatures is shown in Figure 5. The hydrogen permeability coefficient of all materials increases with the temperature rising. The hydrogen permeation at a temperature of 85 °C will be an order of magnitude greater than at room temperature. Relative reductions in permeation at a low temperature are somewhat greater: at −40 °C, permeation is about 2% of the value at room temperature.

Yu Sun [41] tested the hydrogen permeability of PA6 and LIC/PA6 materials at different temperatures (−10 °C, 25 °C, 85 °C) and different pressures (25–50 MPa). The relationship between the hydrogen permeability coefficient and the temperature of the two materials is shown in Figure 6. The logarithm of the permeability coefficient has a linear relationship with the reciprocal of temperature, indicating that the hydrogen permeability coefficient follows Arrhenius’ law with the change in temperature. Zhang and Rogers et al. [36,61] conducted experimental research on the influence of temperature on the hydrogen permeation performance of HDPE and reached the same conclusion.

The hydrogen permeability coefficient of the material increases with the increase in temperature. This is because the rate of gas penetration is closely related to the motion state of the polymer. The free volume of the polymer and the segmental flow of the polymer determine the hydrogen permeability through the polymer. The temperature rise increases the kinetic energy of the polymer chain and activates the volume expansion, resulting in a larger free space for the moving elements to move. When the temperature gradually rises, the chain segments are unlocked from the frozen state, and mutual movement between side groups, branch chains, or links begins to occur, thereby making it easier for hydrogen molecules to penetrate the polymer [41].

In summary, existing research results indicate that the temperature-dependent hydrogen transport characteristics have mainly been studied at elevated temperatures, with limited data available over a broader temperature range, particularly for polymers in the −40 to 85 °C range. As described in Section 3.4, the current standards only require hydrogen permeation tests at normal and high temperatures, and they do not address hydrogen permeation in the low-temperature environment of −40 °C. Extremely low temperatures can increase material brittleness and reduce mechanical properties [64]. Additionally, micro-cracks in the material may more easily expand under the combined action of low temperature and fatigue load [65], leading to increased hydrogen permeability and the risk of low-temperature failure in severe cases [66,67]. Therefore, it is necessary to perform experimental research on the hydrogen permeation performance characteristics of polymer liner materials in the low-temperature environment of −40 °C and explore whether the hydrogen permeation law that governs liner materials under low-temperature conditions is the same as the general law that governs gas permeability for existing polymer materials. 

### 4.2. Test Pressure

In early studies on gas permeability in pressurized environments, Henry et al. [68] analyzed the permeability of inorganic and organic gases in PEs and PPs pipeline materials at pressurization conditions of up to 1.4 MPa. The results showed that the logarithm of the permeability coefficient was linear with the pressure, and the relationship of pressure to permeability rate varied with the gas species. Li [69] also reached the same conclusion at 10 MPa, where Henry’s law was applicable to the permeability coefficient below the critical pressure, but significant deviations occurred above the critical pressure. Stern [70] studied the influence of gas pressure on permeability for inorganic gases (hydroolefins and fluoroolefin-based organic gases) up to 5.5 MPa in semi-crystalline polymers and discussed the relationship between the permeability process and free volume. The change in the permeability coefficient with pressure was attributed to the correlation effect between the increase in free volume caused by osmotic absorption and the decrease in free volume caused by hydrostatic compression. Naito et al. [60] measured the permeability coefficients of LDPE and PP for several gases at a pressure of 8 MPa, and they found that the hydrogen permeability remained almost unchanged. They considered the effects of gas molecule size and solubility on diffusion, separating the pressure dependence of permeability into two factors related to hydrostatic pressure and concentration. However, as described in Section 3.4, the maximum allowable working pressure for 70 MPa type-IV on-board hydrogen storage cylinders is 87.5 MPa, which is nearly 10 times greater than the pressure used in the previous experiments. Therefore, the influence of the high-pressure hydrogen environment on the hydrogen permeability of type IV on-board hydrogen storage cylinders should be considered, including aspects such as material plasticization caused by gas dissolution and the potential impact of crystallinity changes.

In order to study polymer hydrogen permeability under high-pressure conditions, Fujiwara et al. [46] used the HPHP method to analyze the effects of various pressures (10–90 MPa) on the hydrogen permeability of HDPE, as shown in Figure 7. The experimental results showed that the extent of hydrogen permeation increased with pressure, although the rate of increase gradually decreased. Furthermore, the permeability, diffusion, and solubility coefficients all decreased as the pressure increased. The diffusion coefficient decreased more quickly than the solubility coefficient, indicating that at a certain temperature, hydrogen permeation was mainly affected by diffusion rather than solubility. This finding contrasts with the conclusions drawn from experiments at low pressures. Additionally, the gas permeation process under high pressure did not adhere to Henry’s law. Li [69] arrived at the same conclusion. Barth et al. [34] analyzed that the deviation observed under high pressure was due to the hydrostatic effect. The hydrostatic effect reduces the free volume in the polymer, inhibits the diffusion process, and, consequently, reduces the permeability coefficient.

Yu Sun [41] tested the hydrogen permeability of PA6 and LIC/PA6 materials at different pressures (25 MPa, 35 MPa, 50 MPa). The relationship between the hydrogen permeability coefficient and pressure of the two materials is shown in Figure 8. It can be observed that as the pressure increased, the hydrogen permeability coefficient decreased. This phenomenon is attributed to the type of gas molecule involved in the process. For organic vapor and soluble gases such as CO_2_, the permeability coefficient tends to increase as pressure rises, whereas, for sparingly soluble gases such as helium and hydrogen, the permeability coefficient decreases with increasing pressure. This occurs because higher pressures cause the polymer to become compacted, leading to increased density, reduced free volume inside the material, and hindered hydrogen diffusion within it [71,72]. Moreover, Fumitoshi et al. [73] found that the crystallinity of polyethylene increases under a high-pressure hydrogen environment, further impeding hydrogen diffusion. However, this change in crystallinity is reversible, with the polymer’s crystallinity returning to its original value when the pressure returns to atmospheric levels.

As described in Section 3.4, the hydrogen permeation test pressure should be determined in combination with the test temperature and actual on-board hydrogen storage cylinder working conditions. The normal working pressure in the type IV hydrogen storage cylinder is as high as 87.5 MPa. Most of the existing experimental studies are carried out under low pressure, and the experimental data on the hydrogen permeability of polymer materials under high pressure are insufficient. Therefore, further research is needed to better understand the laws governing the hydrogen permeability of polymer liner materials under both high and low-pressure conditions.

Additionally, it is important to consider the operating temperature range of type IV hydrogen storage cylinders, which typically spans from −40 °C to 85 °C, and the working pressure range of 0.1 MPa to 1.25 NWP. These cylinders are subject to alternating pressure loads and continuous temperature changes during service, which can potentially affect the hydrogen permeability of the polymer liner material. In extreme temperature conditions, the liner may be at risk of damage, such as blistering and collapse [18], necessitating higher requirements for the hydrogen permeability resistance of liner materials. Therefore, the effect of alternating pressure load (repeated charging and discharging or rapid pressure relief) and temperature change in the hydrogen permeability characteristics of the polymer liner material of type IV on-board hydrogen storage cylinder are also the issues to be further studied in the future. 

### 4.3. Polymer Material Properties

As described in Section 3.1, the influence of the molding process and weld defects of test samples on the hydrogen permeation test is related to polymer structure and properties. This section discussed the effect of material properties on hydrogen permeability, such as crystallinity, molecular chain, and additives.

In order to understand the hydrogen permeability mechanism, Kotono Takeuchi et al. [30] measured the gas permeability of different equivalent weight (EW) perfluorosulfonic acid (PFSA) membranes. For the high-EW (EW > 909) membranes, higher crystallinity decreased the void fraction in the structure and inhibits the hydrogen diffusion across the aligned polymer chains. This effectively reduced hydrogen solubility and diffusivity, thereby reducing hydrogen permeability. Yersak et al. [19] found that although the crystallinity of polyamide was generally lower than polyethylene, the hydrogen penetration rate of polyamide was lower than that of polyethylene. This indicated that there are many factors affecting the hydrogen permeability of polymer materials, which may also be related to the activation energy density in the amorphous region of polymer materials. Hydrogen bonds can be formed in polyamide, which enhances the interaction between molecular chains and reduces permeability. Pepin et al. [74] found that the hydrogen permeability coefficient of PA12 was five times that of PA6 due to the fact that the hydrogen bond density in PA12 was twice that of PA6. Moreover, the side chain groups of the molecular chain in the polymer and the orientation of the molecular chain also influence the gas permeability to a certain extent. The former affects the movement of molecular chains in the polymer, while the latter is related to the way the polymer is formed [40].

The molding process of polymer materials has some influence on hydrogen permeability. Barton Smith [75] tested the hydrogen permeability coefficient of polymer materials such as HDPE, PA, polyethylene terephthalate (PET), and thermotropic liquid crystal polymer (TLCP), and the test results are shown in Table 6. The result showed that there were no statistically significant differences between the permeability of HDPE produced by extrusion molding, rotational molding, and injection molding manufacturing processes. The permeability of PET was about an order of magnitude less than that of HDPE. The permeability of TLCP is comparable to that of blow-molded PET. Moreover, Condé-Wolter et al. [76] found that there were micro-cracks and other defects in the weld of the liner material, which would affect the mechanical properties and increase the hydrogen permeability coefficient of the liner.

In order to improve the hydrogen barrier properties of the liner, fillers can be added to the polymer. Fillers will form a physical barrier in the polymer, making the hydrogen transport path complicated and increasing the tortuous path of the penetrating molecule into the polymer matrix, as shown in Figure 9. Yu Sun [41] found that after PA6 was added with fillers, the cross path of permeating molecules was increased, thus reducing the rate of hydrogen permeation. In addition, the introduction of filler also increased the stiffness of the material, and the flexibility of the polymer chain decreased, making it more difficult for hydrogen molecules to penetrate the polymer. Therefore, how to prepare the inner liner material with good hydrogen barrier properties will be the future research direction.

## 5. Conclusions and Prospects

The type IV hydrogen storage cylinder has become a research hotspot due to its advantages of light-weight, high-hydrogen storage density, and good fatigue performance. However, hydrogen permeation will occur inevitably in the polymer liner. Therefore, it is necessary and challenging to reduce the hydrogen permeation amount of liner materials as much as possible and to accurately measure the hydrogen permeability rate, which will remain the focus of future research. In this paper, the hydrogen permeation mechanism of the liner material is first discussed in detail. Then, the hydrogen permeation test methods defined by various standards are compared and analyzed, and the factors that affect the process of hydrogen permeation are discussed. This can improve the understanding of the hydrogen permeation mechanism and provide references for subsequent researchers and research on the safety of the liner material of type IV on-board hydrogen storage cylinders. 

So far, several standards have been promulgated by international organizations to require permeation tests of the liner material of type IV on-board hydrogen storage cylinders. From the perspective of on-board hydrogen storage cylinder safety, there are still many issues that need to be further studied:

(1) The welded liner prepared by injection molding may have defects such as wrinkles, creases, and gas cavities near the weld, which will affect the hydrogen permeability at the welding seam of the liner. In terms of sampling position, injection-molded liners with welds may have defects such as wrinkles, creases, and gas cavities near the weld, which can affect the hydrogen permeability at the welding seam of the liner. Therefore, it is necessary to study the performance at the welding seam to ensure that the plastic liner hydrogen permeation test considers the area with the weakest hydrogen permeability. The hydrogen permeation test should consider the influence of the molding process and weld defects of the liner material and determine appropriate sample preparation requirements and sampling position;

(2) It is recommended to keep the drying temperature below the softening temperature of the liner material to prevent softening from affecting test result consistency. However, since on-board hydrogen storage cylinders may operate in high-temperature, high-pressure, and high-humidity environments, it is essential to study the comprehensive influence of humidity, coupled with hydrogen temperature and pressure, on the hydrogen permeability of the liner material to ensure that the hydrogen permeation performance of the liner material meets applicable requirements under high-humidity conditions;

(3) The CSA ANSI CHMC 2 and ISO 11114-5 standards define test devices with similar principles and structures, and all use the HPHP method. However, this method requires a higher sealing effect for the O rings. The effect of the O ring hydrogen permeability coefficient cannot be accurately judged due to the existing test device being sealed via sealing rings. Additionally, since the sample diameter is larger than that of the sealing rings, hydrogen may escape from the sample edge. Hence, it is necessary to optimize the structure of the existing test device to improve hydrogen permeation test accuracy;

(4) The existing standard requires only that the polymer liner material undergo hydrogen permeation tests at normal and high temperatures. Hydrogen permeation tests in a low-temperature environment of −40 °C are not considered. Under the combined action of low temperature and fatigue load, micro-cracks in the material may expand more easily, leading to a decline in the hydrogen permeability of the material. Low-temperature failure may occur in severe cases. Therefore, it is necessary to perform experimental research on the hydrogen permeation performance characteristics of polymer liner materials in the low-temperature environment of −40 °C and to explore whether the hydrogen permeation law for polymer liner materials under low-temperature conditions is the same as the general law that governs gas permeability among existing polymer materials;

(5) The current standards only require hydrogen permeation tests at normal and high temperatures, and they do not address hydrogen permeation in the low-temperature environment of −40 °C. Micro-cracks in the material may more easily expand under the combined action of low temperature and fatigue load, leading to increased hydrogen permeability and the risk of low-temperature failure in severe cases. The normal working pressure in the type IV hydrogen storage cylinder is as high as 87.5 MPa. Most of the existing experimental studies are carried out under low pressure, and the experimental data on the hydrogen permeability of polymer materials under high pressure are insufficient. Therefore, further research is needed to better understand the laws governing the hydrogen permeability of polymer liner materials under both high and low-pressure conditions.

Hydrogen permeation test methods in existing standards are different in sample preparation, sample pretreatment, test device, test temperature and pressure, and qualification indicators. The existing hydrogen permeation test method needs to be further improved. In addition, the operating temperature range of the type IV hydrogen storage cylinder is between −40 °C and 85 °C, and the working pressure range is between 0.1 MPa and 1.25 NWP. Since the cylinder is often affected by alternating pressure loads and continuous temperature changes during service, which can potentially affect the hydrogen permeability of the polymer liner material. This puts forward higher requirements for the hydrogen permeability resistance of liner materials. Therefore, the effect of alternating pressure load (repeated charging and discharging or rapid pressure relief) and temperature change on the hydrogen permeability characteristics of the polymer liner material of type IV on-board hydrogen storage cylinder are also the issues to be further studied in the future. 

## Figures and Tables

**Figure 1 materials-16-05366-f001:**
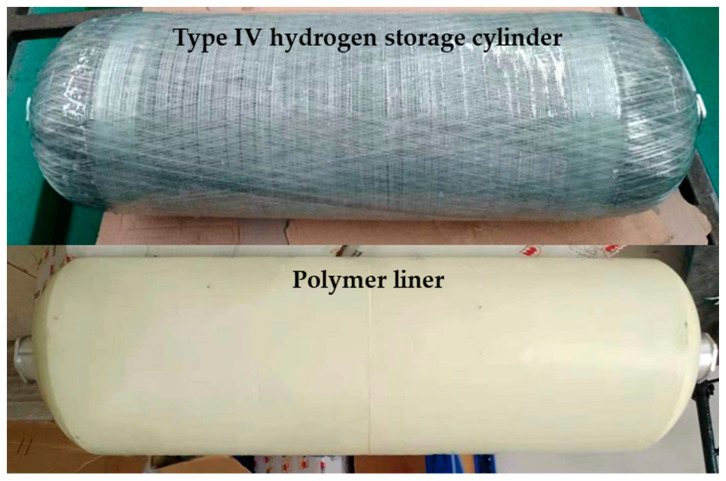
The structure of the type IV hydrogen storage cylinder and polymer liner.

**Figure 2 materials-16-05366-f002:**
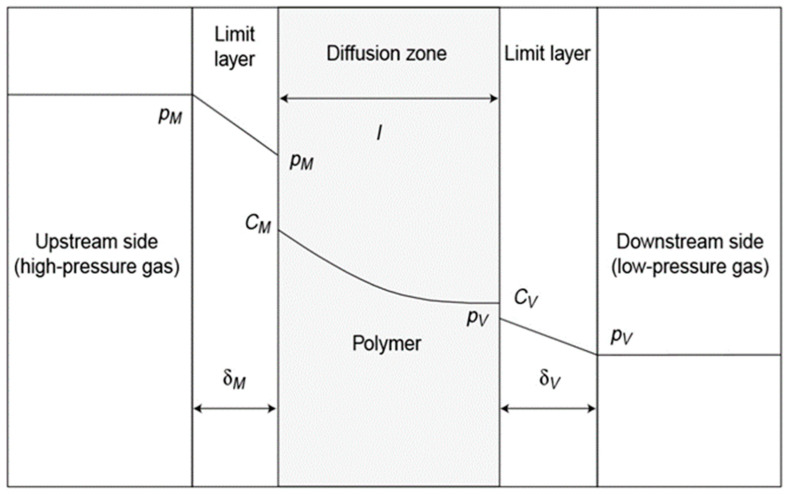
The diffusion process of gas in the polymer, reproduced with permission from [38].

**Figure 3 materials-16-05366-f003:**
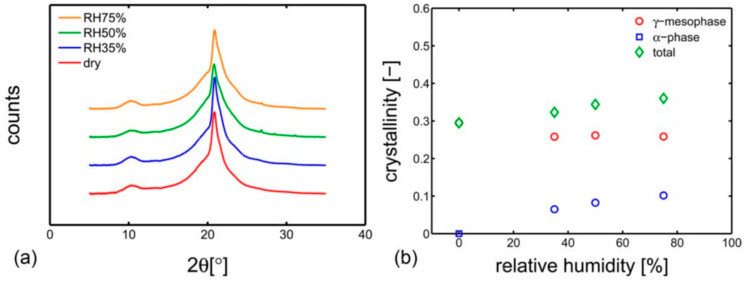
(**a**) Wide angle X-ray diffraction integrated patterns of samples conditioned at different relative humidity and 23 °C; (**b**) crystalline fractions as functions of relative humidity, obtained by deconvolution analysis, reproduced with permission from [48].

**Figure 4 materials-16-05366-f004:**
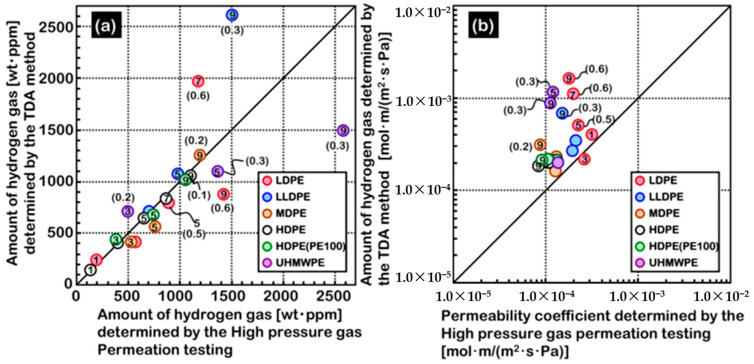
Correlativity of HPHP and TDA method. (**a**) Amount of penetrated hydrogen gas. (**b**) Permeability. Numbers in the plot show the exposed pressure, reproduced with permission from [53].

**Figure 5 materials-16-05366-f005:**
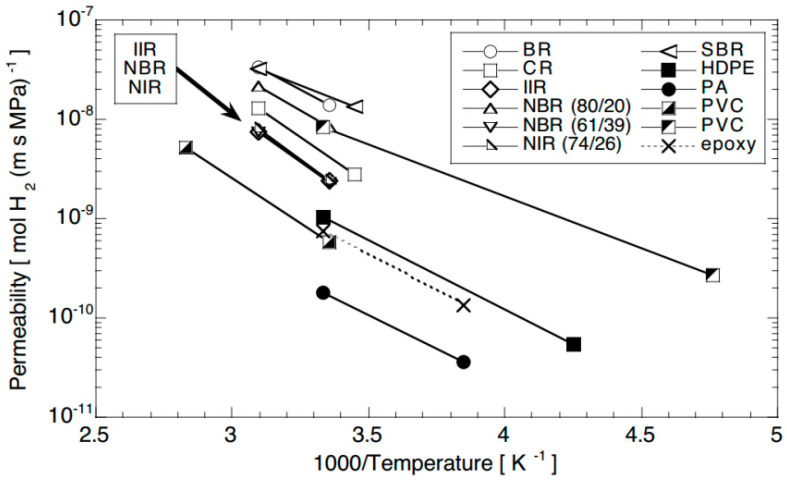
Temperature dependence of hydrogen permeability of several polymer materials, reproduced with permission from [34].

**Figure 6 materials-16-05366-f006:**
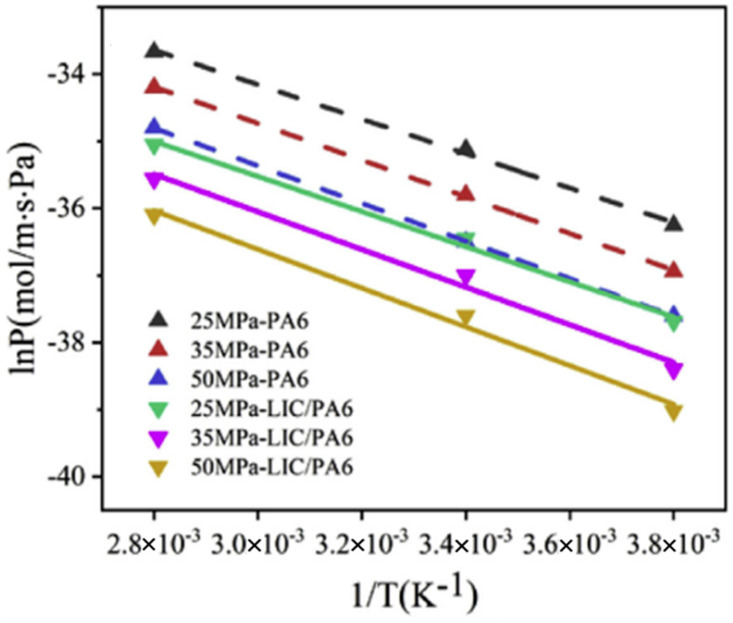
The relationship between hydrogen permeability coefficient and temperature of PA6 and LIC/PA6, reproduced with permission from [41].

**Figure 7 materials-16-05366-f007:**
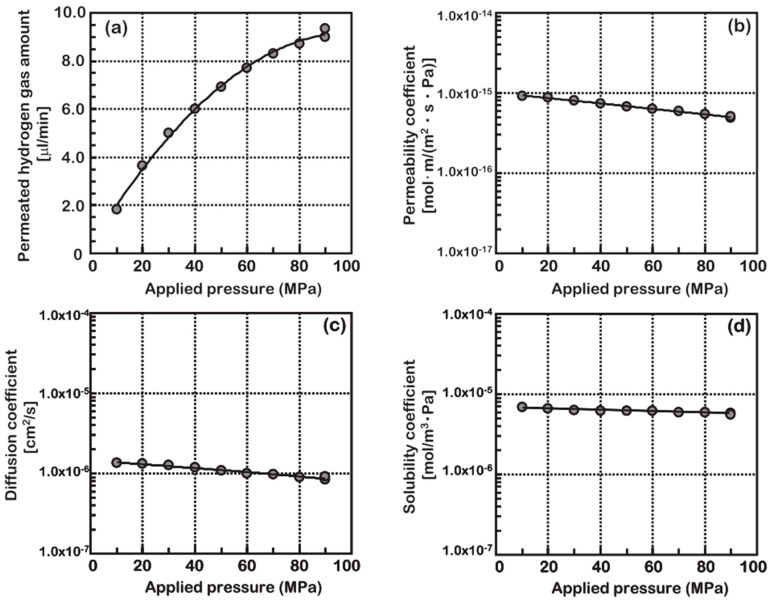
The results of high-pressure permeation tests performed at 30 °C. The influence of the applied pressure on the (**a**) amount of hydrogen gas that permeates, (**b**) permeability, (**c**) diffusion coefficient, and (**d**) solubility coefficient, reproduced with permission from [46].

**Figure 8 materials-16-05366-f008:**
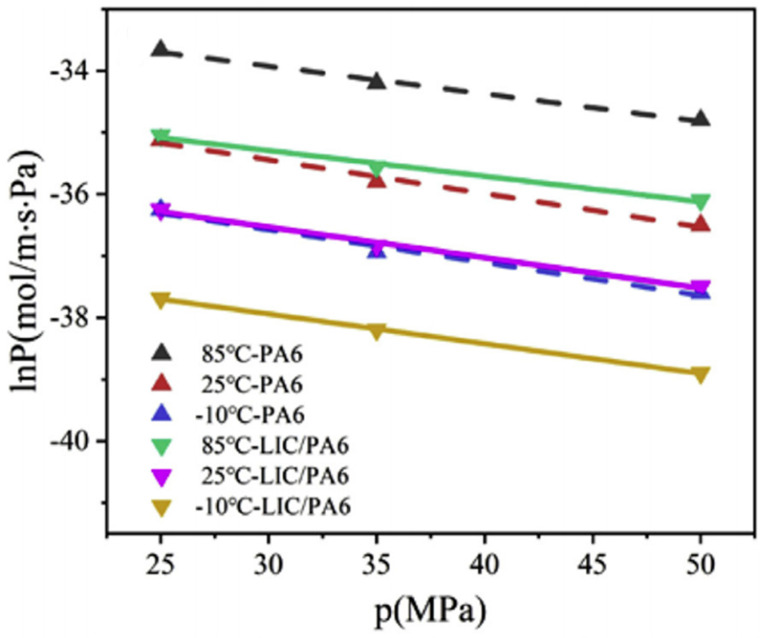
The relationship between hydrogen permeability coefficient and pressure of PA6 and LIC/PA6, reproduced with permission from [41].

**Figure 9 materials-16-05366-f009:**
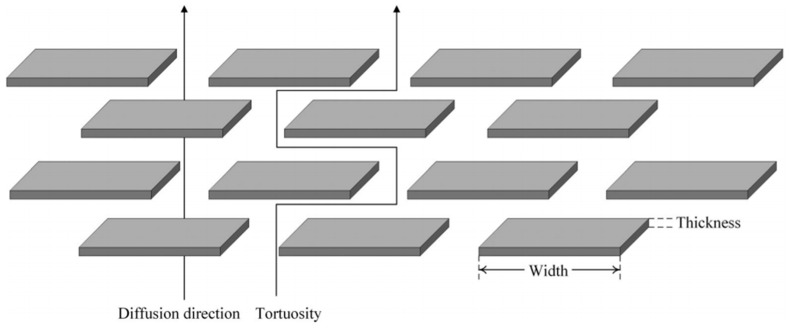
Increased tortuous path length, generated by orthogonally arranged two-dimensional obstacles, reproduced with permission from [35].

**Table 1 materials-16-05366-t001:** Comparison of the sample preparation requirements imposed by various standards.

Standard	CSA ANSI CHMC 2	ISO 11114-5	T/CATSI 02 007
Sample source	Sample from polymer liner or prepared from polymer liner produced using same molding process as the product	Same as CSA ANSI CHMC 2	Sample from polymer liner
Sampling position	Unspecified	Unspecified	Seamless liner: middle of cylinder Welded liner: middle of cylinder away from the welding seam
Diameter (mm)	≥25 mm (recommend 78)	Between 40 mm and 80 mm	78 mm
Thickness (mm)	1 ≤ liner thickness ≤ 6 (recommend liner thickness)	Liner thickness	Liner thickness

**Table 2 materials-16-05366-t002:** Comparison of sample pretreatment requirements imposed by various standards.

Standard	CSA ANSI CHMC 2	ISO 11114-5	T/CATSI 02 007
Drying temperature	Recommend 60 °C	Recommend 65 °C	Unspecified
Vacuum pressure	Unspecified	Recommend 10–50 mbar	Unspecified
Vacuum drying finish conditions	Mass loss rate within 48 h or 1 h of drying is <0.5%	The mass loss rate within 24 h of drying is <0.1%	Unspecified

**Table 3 materials-16-05366-t003:** Comparison of the test devices used by the different standards.

Standard	CSA ANSI CHMC 2	ISO 11114-5	T/CATSI 02 007
Device structure diagram	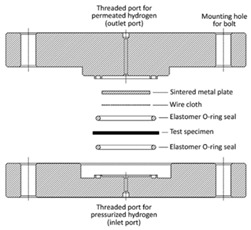	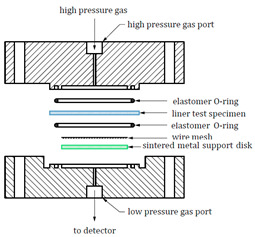	Unspecified
Test Method	HPHP method	HPHP method	HPHP method
Sintered metal plate porosity	Porosity reaches grade 2	Unspecified	Unspecified
Wire mesh size	140 μm	150 μm	Unspecified
Hydrogen exposure zone diameter	≥di mm	≥di mm	≥di mm

**Table 4 materials-16-05366-t004:** Comparison of test temperatures and pressures.

Standard	CSA ANSI CHMC 2	ISO 11114-5	T/CATSI 02 007
Test temperature and pressure	(85 ± 1) °C, 1.25 NWP, (15 ± 1) °C, 1.25 NWP, (85 ± 1) °C, 0.1 NWP, (15 ± 1) °C, 0.1 NWP.	Test should be carried out at a certain pressure and temperature	(55 ± 1) °C, 1.15 NWP, (15 ± 1) °C, NWP, (55 ± 1) °C, 0.1 MPa, (15 ± 1) °C, 0.1 MPa.

**Table 5 materials-16-05366-t005:** Comparison of qualification indicators defined by various standards.

Standard	CSA ANSI CHMC 2	ISO 11114-5	T/CATSI 02 007
Qualification indicators	The steady-state gas transmission rate measured at 15 °C for a sample with a diameter of 78 mm is divided into six rating values: 10: ≤0.8 Ncm³/h, 8: >0.8–1.5 Ncm³/h, 6: >1.5–3 Ncm³/h, 4: >3–6 Ncm³/h, 2: >6–16 Ncm³/h, and 0: >16 Ncm³/h.	Unspecified	At 15 °C, *P_e_* ≤ 1 × 10^−13^ cm^3^·cm/(cm^2^·s·Pa), At 55 °C, *P_e_* ≤ 1 × 10^−12^ cm^3^·cm/(cm^2^·s·Pa)

**Table 6 materials-16-05366-t006:** Test results of hydrogen permeability coefficient of different polymer materials under different molding processes.

Polymer	Pressure/Bar	*P_e_*/mol H_2_·m/(m^2^·s)
Injection-molded HDPE	134	2.08 × 10^−10^
Extrusion-molded HDPE	104	5.51 × 10^−11^
Rotomolded HDPE	345	2.58 × 10^−10^
Extrusion-molded PA6	137	3.38 × 10^−11^
Compression-molded TLCP	136	7.09 × 10^−13^
Blow-molded PET	137	6.71 × 10^−12^

## Data Availability

Not applicable.

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
