# Peer review of "Review of the Hydrogen Permeation Test of the Polymer Liner Material of Type IV On-Board Hydrogen Storage Cylinders"

_materials, 2023, doi:10.3390/ma16155366_

Round 1

Reviewer 1 Report (New Reviewer)

The work is focused on information on hydrogen permeation tests procedures, for the polymer material used in liner of Type IV hydrogen storage cylinders.

The introduction section describes the context of the technologies and applications, where the compressed H2 cylinders - and therefore their liners - are used, in quite a concise way, and then proceed to list all the appliable standards with a brief explanation of the procedures. The section quotes these international and national standards, but it contains also references to a sufficient number of papers from scientific literature.

The second section covers the basics of hydrogen (or gases, in general) permeation/diffusion in a solid layer and the mechanism behind this phenomenon. It cites and reports the fundamental laws of diffusion, allowing a simplified understanding of the physics behind the following parts.

The following part reviews and evaluates the hydrogen permeation test procedures for polymer liner materials, as specified by the several standards already mentioned before. This section takes into account and discuss the impact of various elements (including pre-treatment, preparation, test machine, temperature and pressure) on the outcomes of the evaluation. There are some suggestions as well, meant to enhance the test procedure. This part is lacking some deeper insight about the effects of the discussed factors in the test results: since the work under revision is a Review, one would expect a more effective survey of scientific works about the subject in relation with the properties of the specific polymer; if hydrogen diffusion is not so much discussed, there could be some suggestions or theories related to the phenomenon with other gases/material. In my opinion writing: «However, there are few researches on the influence of permeation properties and hydrogen permeation test methods of the polymer liner material of on-board type IV hydrogen storage cylinder.» is not a valid motivation to avoid a correlation between the requirements found in the standards and the scientific research in the field. For example, it is my understanding that the weld area should be inspected more and surely the process of blow molding has been intensively investigated in polymer research, evidencing defects and flaws in the welding area. Some references to previous studies could corroborate the idea presented.

In the discussion, through a scarce review of the literature, the impact of variables like the test temperature and pressure on the hydrogen permeation process is evaluated. Concerning the temperature, two cases are investigated: high temperatures, increasing diffusivity following Arrhenius’ law, and low temperatures, influencing the mechanical properties of the polymers and therefore increasing risks of failure. In both cases, in my opinion, references to literature and cross-linking of material properties with effects are insufficient for a paper that is a review and should provide the readers with detailed information and many sources. The subsection dealing with pressure is also quite concise, and lacks an original contribution in connecting references and the objective of the standard procedures.

The conclusion section is quite nicely arranged and sums up all the main concepts and themes discussed throughout the paper, underlining all the missing details and the points where both standard procedures and the research state-of-the-art are lacking some improvements and deeper understanding. This is a nice idea, however the “suggestions” for future research and prospective are quite elementary in the section and they could use a bit more connections with the hypothesis advanced in the previous sections.

Finally, I would recommend the authors to increase the value of the work by quoting and referencing many more papers and experimental works (not only related to H2 permeation and standards for Type IV cylinders, but also basic researches on diffusivity in polymers and properties thereof) and especially to connect the knowledge provided by these papers with the requirements of standards. This is only partially done and the resulting list of requirements and possible improvements is not so appealing as it is presented right now. Therefore, I find the idea and the structure of the work promising, but I recommend intensive revisions and further literature survey before publication.

The English language does not require major editing, just some minor review; with the exception of the abstract, which looks like written from a different author. I suggest to correct this part with more care, because is the way the paper is presented to the audience.

Author Response

Reviewer 2 Report (New Reviewer)

Major Revision

1. Title very long

2. Add more references in the introduction to emphisize the state of the arts of this matter

3. more Figures

4. cite your equations on the text

5. cite your previous work too

7. references need check by follwing the journal template and add the 2023 ones

with regards

Round 2

Reviewer 2 Report (New Reviewer)

Accepted as it is in the revised format

This manuscript is a resubmission of an earlier submission. The following is a list of the peer review reports and author responses from that submission.

Round 1

Reviewer 1 Report

The presented article is devoted to the actual topic of storage of compressed hydrogen. The article discusses in detail the current standards for certification of the validity of cylinders with polymer liners. In this case, the main suitability parameter is the hydrogen permeability of the polymer lining.

The manuscript presents in sufficient detail the mechanism of hydrogen diffusion in a solid medium, the sample preparation technique, and test conditions in the developed standards. Recommendations are given for the necessary refinement of existing standards.

The introduction to the manuscript should also mention other technologies for storing and transporting hydrogen (liquefied, LLHC, etc.) with appropriate references. What are the gas phase (hydrogen) requirements for standard testing? Hydrogen permeability can be affected by trace impurities of methane (or other hydrocarbons), nitrogen (purified hydrogen after ammonia decomposition) and carbon dioxide (purified hydrogen after steam reforming of methane).

The article can be accepted in the "Materials" journal with minor revision.

Minor editing of English language required

Reviewer 2 Report

Occasional missing spaces in the standards names were found in the manuscript.

Reviewer 3 Report

Authors studied the hydrogen permeation mechanism of the liner material. They defined hydrogen permeation test methods by various standards which they compared and analyzed. They studied  sample preparation; sample pretreatment; the test device; the test temperature and pressure; qualification indicators, etc. Furthermore, factors such as the temperature and pressure that affect the hydrogen permeation process are summarized.  Overall, the work is good and it is suitable for the publication in Materials Journal provided the following minor comments need to be addressed first.    -Abstract needs to be re-written. First, a few general sentences can be moved to the introduction section so that more technical details can be included. -The work is related to hydrogen storage and in this regard, the following latest relevant literature can be discussed in it to strengthen the literature review: Carbon 207, 23-35, 2023. -Authors mentioned that "gas transmission 296 rate of the plate is at least 100 times greater than that". Please cite appropriate ref.  -Use lower - and upper-case letters in the figure captions uniformly. Typo in caption of Fig. 1.